# Active RIS-Assisted Uplink NOMA with MADDPG for Remote State Estimation in Wireless Sensor Networks

**DOI:** 10.3390/s25154878

**Published:** 2025-08-07

**Authors:** Rongzhen Li, Lei Xu

**Affiliations:** School of Computer Science and Engineering, Nanjing University of Science and Technology, Nanjing 210094, China; lirongzhen0213@163.com

**Keywords:** wireless sensor networks, uplink NOMA, RIS, MADDPG algorithm, RSE error

## Abstract

Non-orthogonal multiple access (NOMA) and reconfigurable intelligent surfaces (RISs) are recognized as key technologies for beyond 5G and 6G wireless communications. To address the high computational complexity and non-convex optimization challenges, this letter proposes an optimization framework based on the Multi-Agent Deep Deterministic Policy Gradient (MADDPG) algorithm. The proposed framework jointly makes use of sensor grouping, power allocation, an RIS computation strategy, and phase shifts to minimize the remote state estimation (RSE) error. Simulation results demonstrate that the MADDPG algorithm, when applied in an RIS-assisted NOMA system, significantly reduces the RSE error.

## 1. Introduction

With the increasing number of wireless devices and the development of ultra-5G and 6G technologies in industrial Internet of Things (IoT) networks, wireless sensor networks (WSNs) [1], as a key component of IoT systems, face higher demands for low latency, low remote state estimation (RSE) errors, and low system error rates (SERs) [2,3,4]. As the number of smart devices grows and large-scale IoT applications proliferate, ensuring the efficiency and reliability of these systems has become a critical issue to address.

To enhance spectrum efficiency in WSNs, non-orthogonal multiple access (NOMA) technology has been widely adopted due to its capability to support multiple sensors with the same time–frequency resources through superposition coding and successive interference cancellation [5,6]. Recent studies have enhanced NOMA further by integrating active reconfigurable intelligent surfaces (RISs) to boost the link reliability and energy efficiency. For instance, the work presented in [7] investigates joint UAV trajectory optimization and active RIS control in NOMA systems, highlighting significant energy efficiency gains. However, this study primarily addressed aerial–terrestrial network scenarios and was not directly applicable to ground-based sensor networks.

Both passive and active RIS technologies have been explored extensively for improving NOMA system performance. Recent research, such as that presented in [8], demonstrates passive RISs’ effectiveness in reducing the bit error rate (BER) and extending the network coverage. Nevertheless, a passive RIS lacks active amplification capabilities and adaptability under highly dynamic channel conditions. Conversely, an active RIS, as explored in [9], enhances the physical-layer security by providing analytical derivations for the secrecy outage probability. However, this study does not incorporate critical aspects such as sensor grouping strategies or estimation error optimization.

Traditional optimization methods like block coordinate descent (BCD) and successive convex approximation (SCA) have been employed for resource allocation in RIS-assisted NOMA networks. For example, the research by Xu et al. [10] achieves improved system sum rates using BCD-based resource scheduling techniques. Additionally, recent efforts such as [11] have explored low-complexity signal processing schemes, including sparse channel parameter estimation for MIMO-FBMC systems, which are highly relevant to industrial big data communication scenarios. However, these approaches typically focus on the physical layer and do not address joint decision-making strategies or cross-layer coordination for sensor estimation. Moreover, they lack multi-agent coordination mechanisms and considerations for remote state estimation (RSE) error control, which are essential in practical WSN deployments.

Meanwhile, reinforcement learning methodologies, including deep reinforcement learning (DRL), deep deterministic policy gradient (DDPG), and multi-agent DDPG (MADDPG), have emerged as effective solutions for dynamic resource allocation. A recent IEEE Transactions study [12] presents an attention-augmented MADDPG framework applied to NOMA-based vehicular mobile edge computing, demonstrating improved convergence rates and predictive accuracy under realistic workload scenarios. Additionally, recent studies such as [13] have employed DRL approaches to jointly optimize the UAV trajectories, transmission power, and RIS phase shifts in RIS-aided UAV-NOMA systems. Nonetheless, these studies have predominantly considered passive RIS configurations and assumed static user grouping, limiting the adaptability in dynamic scenarios.

To the best of our knowledge, no existing research has jointly tackled the sensor state estimation error (RSE), sensor grouping, transmission power control, and active RIS beamforming within an uplink NOMA-assisted WSN using a decentralized multi-agent reinforcement learning framework. To this end, we propose a MADDPG-based distributed policy learning approach. MADDPG is particularly suitable for this problem setting because it explicitly models the interactions among agents and handles decentralized control in partially observable environments. In our scenario, each sensor acts as an agent, making decisions based only on its local state, such as channel gain and estimation covariance. This naturally forms a Dec-POMDP (Decentralized Partially Observable Markov Decision Process), to which MADDPG is well suited. It enables coordinated learning during centralized training while maintaining independent execution, supports hybrid discrete–continuous action spaces, and offers scalability in large sensor networks with dynamic channel conditions.

The main contributions of this paper are summarized as follows:1.Joint Integration of an Active RIS and MADDPG: We propose a joint optimization framework that integrates an active RIS and MADDPG-based decentralized learning for uplink WSNs. This design improves the estimation accuracy, reduces the power consumption, and enhances the system’s adaptability.2.Real-Time Estimation via Kalman Filtering: The proposed method incorporates Kalman filtering (KF) to enable real-time state tracking, which improves the estimation robustness under varying network conditions.3.Decentralized Resource Allocation: We formulate a novel decentralized resource allocation problem that jointly considers sensor grouping, power control, and active RIS beamforming to minimize the RSE error—an optimization objective rarely addressed in existing works. Each sensor acts as an independent agent and makes decisions based on its local state in a Dec-POMDP setting. The MADDPG framework is employed to enable scalable and coordinated learning among agents. While our method is built upon an established RL architecture, it is tailored to a unique estimation-driven optimization task in WSNs. We also acknowledge the absence of a formal theoretical convergence analysis and highlight this as a direction for future investigation.4.Performance Gains Verified by Simulations: Simulation results show significant improvements in the RSE error reduction and system reliability compared to those under conventional and single-agent approaches, confirming the practicality of the proposed framework.

The remainder of this paper is organized as follows: Section 2 details the local sensor state estimation and the uplink NOMA transmission model with an active RIS, including the derivation of the RSE error metrics and the outage probability model. Section 3 introduces the MADDPG-based decentralized algorithm, elaborating on the joint optimization framework that integrates sensor grouping, power allocation, and active RIS beamforming, as well as incorporating KF or enhanced real-time state estimation. Section 4 presents the simulation experiments, performance comparisons with benchmark algorithms, and a comprehensive analysis of the results demonstrating improvements in the RSE accuracy and network reliability. Finally, Section 5 concludes this paper by summarizing the key findings and potential directions for future research.

Notations: R denotes the set of real numbers. C represents the set of complex numbers. The symbol ${\left(\cdot \right)}^{T}$ denotes the transpose operation, and the superscript ${\left(\cdot \right)}^{-1}$ indicates the matrix inverse. The notation P[·] refers to probability and E[·] to expectation. Tr(·) represents the trace operator of a matrix. A bold symbol (e.g., P) is used to denote vectors or matrices, with lowercase bold letters representing vectors and uppercase bold letters representing matrices. Non-bold symbols (e.g., *p*) indicate scalar quantities such as real-valued parameters or constants.

## 2. The System Model

This letter explores an RSE system within an uplink NOMA transmission model, aided by an active RIS. As depicted in Figure 1, the system consists of a single BS, equipped with *G* receive antennas, an active RIS with *M* reflective elements and *N* single-antenna sensors. The sets of sensors, sensor groups, and reflective elements are denoted as $N=\left\{1,2,\ldots ,N\right\}$, $G=\left\{1,2,\ldots ,G\right\}$ and $M=\left\{1,2,\ldots ,M\right\}$, respectively.

This letter explores an RSE system within an uplink NOMA transmission model, aided by an active RIS. As depicted in Figure 1, the system consists of a single BS equipped with *G* receive antennas; an active RIS with *M* reflective elements; and *N* single-antenna sensors. To facilitate decentralized decision-making, the *N* sensors are partitioned into *G* sensor groups, each served by the BS equipped with *G* antennas. Note that the grouping does not imply a one-to-one mapping between each group and a specific antenna; instead, it is designed for efficient multi-agent training and resource allocation. The sets of sensors, sensor groups, and reflective elements are denoted as N={1,2,…,N}, G={1,2,…,G}, and M={1,2,…,M}, respectively.

### 2.1. Local State Estimation

The output observed by sensor *n* at time *k* for the associated linear time-invariant (LTI) process can be expressed as follows(1)$$\begin{array}{cc} \; & x_{n}^{k+1}=A_{n}x_{n}^{k}+w_{n}^{k}, \end{array}$$(2)$$\begin{array}{cc} \; & y_{n}^{k+1}=C_{n}x_{n}^{k}+v_{n}^{k}, \end{array}$$
where $x_{n}^{k}\in R^{l_{n}}$ and $y_{n}^{k}\in R^{r_{n}}$ represent the state and measurement vectors, respectively  [14], with $l_{n}$ and $r_{n}$ denoting the dimensions of the state and measurement vectors for sensor *n*. The state evolution is governed by the state transition matrix $A_{n}\in R^{l_{n}\times l_{n}}$, while the measurement process is characterized by the measurement matrix $C_{n}\in R^{r_{n}\times l_{n}}$. The disturbances $w_{n}^{k}\in R^{l_{n}}$ and $v_{n}^{k}\in R^{r_{n}}$ are assumed to be zero-mean, independent, and identically distributed (i.i.d.) Gaussian noise processes, with covariance matrices $Q_{n}$ and $R_{n}$, respectively.

Each sensor operates a local Kalman filter (KF) and transmits the corresponding minimum mean squared error (MMSE) state estimate to the remote estimator, along with the associated remote state estimation (RSE) error covariance $P_{n}^{k|k}$, which are defined as  [15](3)$$\begin{array}{cc} {\hat{x}}_{n}^{k|k} & ≜E\left[x_{n}^{k}|y_{n}^{1},y_{n}^{2},\ldots ,y_{n}^{k}\right], \end{array}$$(4)$$\begin{array}{cc} P_{n}^{k|k} & ≜E\left[\left(x_{n}^{k}-{\hat{x}}_{n}^{k|k}\right){\left(x_{n}^{k}-{\hat{x}}_{n}^{k|k}\right)}^{⊤}\right]. \end{array}$$

The prior state estimate ${\hat{x}}_{n}^{k|k-1}$ and the posterior state estimate ${\hat{x}}_{n}^{k|k}$ at time *k* are updated through the following steps. The Kalman gain is denoted by $K_{n}^{k}$, and the error covariances are represented by $P_{n}^{k|k-1}$ and $P_{n}^{k|k}$, corresponding to the prior and posterior error covariances, respectively. The KF update process is given by(5)$$\begin{array}{cc} {\hat{x}}_{n}^{k|k-1} & =A_{n}{\hat{x}}_{n}^{k-1|k-1}, \end{array}$$(6)$$\begin{array}{cc} P_{n}^{k|k-1} & =A_{n}P_{n}^{k-1|k-1}A_{n}^{⊤}+Q_{n}, \end{array}$$(7)$$\begin{array}{cc} K_{n}^{k} & =P_{n}^{k|k-1}C_{n}^{⊤}{\left(C_{n}P_{n}^{k|k-1}C_{n}^{⊤}+R_{n}\right)}^{-1}, \end{array}$$(8)$$\begin{array}{cc} {\hat{x}}_{n}^{k|k} & ={\hat{x}}_{n}^{k|k-1}+K_{n}^{k}\left(y_{n}^{k}-C_{n}{\hat{x}}_{n}^{k|k-1}\right), \end{array}$$(9)$$\begin{array}{cc} P_{n}^{k|k} & =\left(I_{l_{n}}-K_{n}^{k}C_{n}\right)P_{n}^{k|k-1}, \end{array}$$
where $I_{l_{n}}$ is the $l_{n}\times l_{n}$ identity matrix, and $l_{n}$ denotes the dimension of the state vector $x_{n}^{k}$.

The local estimation error covariance converges exponentially to a steady-state value. Without loss of generality, we assume that each sensor’s KF reaches this steady state. To simplify the following analysis, we denote the steady-state covariance as $P_{n}^{k}={\bar{P}}_{n}$, k⩾1.

### 2.2. The Uplink Communication Model

After obtaining ${\hat{x}}_{n}^{k}$, sensor *n* transmits it to the BS as a data packet. The communication between the sensors and the BS is based on an uplink NOMA communication model, supported by an active RIS. In this model, the RIS serves as an enhancement tool for signal propagation, optimizing both the signal quality and coverage.

The equivalent baseband time-domain channel in the system is composed of three parts: the channel from the RIS to the BS, from sensor *n* to the RIS, and from sensor *n* to the BS. These channels are denoted by $h^{rb}\in C^{G\times M}$, $h_{n}^{sr}\in C^{M\times 1}$, and $h_{n}^{sb}\in C^{G\times 1}$, respectively. The reflection coefficient of the RIS element *m* is expressed as $\theta_{m}=\alpha_{m}e^{j\phi_{m}}$, where $\alpha_{m}$ represents the reflection amplitude and $\phi_{m}\in \left[0,2\pi \right]$ denotes the phase shift. The RIS beamforming is then captured by the reflection coefficient matrix Θ≜ diag$\left\{\theta_{1},\theta_{2},\ldots ,\theta_{M}\right\}$  [2]. Unlike passive RISs, active RISs are equipped with amplifiers that consume additional power, making thermal noise at the active RIS significant. Consequently, the signal received at the BS can be modeled as(10)$$y_{g}^{k}=\sum_{n=1}^{N}b_{n,g}^{k}\sqrt{p_{n}^{k}}h_{n}^{k}f_{n}^{k}+n_{g}^{k},$$
where $b_{n,g}^{k}\in \left\{0,1\right\}$ represents the binary channel selection for sensor *n*. The total channel gain $h_{n}^{k}$ is denoted as $h_{k}^{rb}\Theta_{k}h_{n,k}^{sr}+h_{n,k}^{sb}$, $h_{n}^{k}\in C^{G\times 1}$. The noise $n_{g}^{k}$ is defined as $h_{k}^{rb}\Theta_{k}v_{k}+n_{0,g}^{k}$, $n_{g}^{k}\in C^{G\times 1}$. The thermal noise at the RIS is modeled by $v_{k}\in C^{M\times 1}$, which follows a circularly symmetric complex Gaussian distribution, i.e., $v_{k}∼CN\left(0_{M},\sigma_{1}^{2}I_{M}\right)$. Similarly, the thermal noise at the BS is modeled by $n_{0,g}^{k}\in C^{G\times 1}$, where $n_{0,g}^{k}∼CN\left(0_{G},\sigma_{2}^{2}I_{G}\right)$ [2]. Let $f_{n}^{k}$ represent the data symbol transmitted by sensor *n* at time *k*, while $p_{n}^{k}$ denotes the corresponding power used for computation offloading. We assume that the transmitted symbols are independent and identically distributed (i.i.d.) as $f_{n}^{k}∼CN\left(0,1\right)$. Since it is an uplink NOMA channel, we assume that the channel gains between different sensors decrease, i.e., $|h_{1}^{k}{|}^{2}\geq |h_{2}^{k}{|}^{2}\ldots \geq {\left|h_{n}^{k}\right|}^{2}$.

The channel model comprises both path loss and small-scale fading. For the direct link between the BS and the sensor *n*, we assume Rayleigh fading. The corresponding channel can be expressed as(11)$$h_{n,k}^{sb}=\sqrt{d_{S,n,k}^{-\alpha_{S}}}{\tilde{h}}_{n,k}^{sb},$$
where $d_{S,n,k}$ is the distance between the BS and sensor *n*, $\alpha_{S}$ is the path loss exponent, and ${\tilde{h}}_{n,k}^{sb}\in C^{G\times 1}$ represents the small-scale fading channel, modeled as $CN\left(0_{G},I_{G}\right)$.

The active RIS is deployed at a fixed location, where line-of-sight (LoS) links exist from both the BS to the RIS and from the RIS to sensor *n*. Therefore, the reflection channel follows Rician fading. The channel matrix from the RIS to the BS, $h_{k}^{rb}$, and the channel vector from sensor *n* to the RIS, $h_{n,k}^{sr}$, are given by [16](12)$$\begin{array}{cc} h_{k}^{rb} & =\sqrt{d_{B,k}^{-\alpha_{B}}}\left(\sqrt{\frac{\delta_{B}}{\delta_{B}+1}}{\tilde{h}}_{k}^{rb,l}+\sqrt{\frac{1}{\delta_{B}+1}}{\tilde{h}}_{k}^{rb}\right), \end{array}$$(13)$$\begin{array}{cc} h_{n,k}^{sr} & =\sqrt{d_{R,n,k}^{-\alpha_{R}}}\left(\sqrt{\frac{\delta_{R}}{\delta_{R}+1}}{\tilde{h}}_{n,k}^{sr,l}+\sqrt{\frac{1}{\delta_{R}+1}}{\tilde{h}}_{n,k}^{sr}\right), \end{array}$$
where $d_{B,k}$ and $d_{R,n,k}$ represent the distances from the BS to the RIS and from the RIS to sensor *n*, respectively. $\alpha_{B}$ and $\alpha_{R}$ are the path loss exponents, while $\delta_{B}$ and $\delta_{R}$ denote the Rician factors. The terms ${\tilde{h}}_{k}^{rb,l}\in C^{G\times M}$ and ${\tilde{h}}_{n,k}^{sr,l}\in C^{M\times 1}$ represent the LoS components from the RIS to the BS and from the sensor to the RIS, respectively. The small-scale fading components, ${\tilde{h}}_{k}^{rb}\in C^{G\times M}$ and ${\tilde{h}}_{n,k}^{sr}\in C^{M\times 1}$, follow the complex Gaussian distributions $CN(0_{G\times M},I_{G\times M})$ and $CN(0_{M},I_{M})$, respectively.

Therefore, according to the uplink NOMA decoding protocol with successive interference cancellation (SIC), the signal from sensor *n* is decoded after those from the sensors with smaller indices (i.e., j<n), whose signals are assumed to have been successfully decoded and removed at the BS. Thus, the residual interference with sensor *n* originates only from the sensors with indices greater than *n*. The signal-to-interference-plus-noise ratio (SINR) of sensor *n* is given by(14)$$\gamma_{n}^{k}=\frac{p_{n}^{k}{∥h_{n}^{k}∥}^{2}}{\sum_{j=n+1}^{N}b_{j,g}^{k}p_{j}^{k}{∥h_{j}^{k}∥}^{2}+\sigma_{k}^{2}},$$
where $\sigma_{k}^{2}≜\sigma_{1,k}^{2}{∥h_{k}^{rb}\Theta_{k}∥}^{2}+\sigma_{2,k}^{2}$ accounts for the effective noise at the BS, including thermal noise from both the RIS and the BS receiver.

In digital communication theory, the SER for a single sensor is expressed in terms of the SINR as $\mathrm{SER}=Q\left(\sqrt{\gamma }\right)$, where $Q\left(x\right)=\frac{1}{\sqrt{2\pi }}\int_{x}^{\infty }e^{-\frac{t^{2}}{2}}\;dt$ denotes the Gaussian Q-function. By applying the above theory, the SER can be approximated as $\mathrm{SER}\approx e^{-\gamma }$ [3]. For sensor *n* at time *k*, the SER is approximated as(15)$${\mathrm{SER}}_{n}^{k}\approx e^{-\gamma_{n}^{k}}.$$

### 2.3. Remote State Eatimation

Each sensor transmits its local state estimation measurement data to the remote estimator (i.e., the BS) via a wireless channel, utilizing uplink NOMA technology enhanced by RIS assistance. Thus, the transmission of ${\hat{x}}_{n}^{k}$ can be characterized by a binary random process $\left\{\lambda_{n}^{k}\right\}$, where (see Figure 2)(16)$$\lambda_{n}^{k}=\left\{\begin{array}{cc} 1, & \mathrm{if}\;{\hat{x}}_{n}^{k}\;\mathrm{arrives}\;\mathrm{error-free}\;\mathrm{at}\;\mathrm{time}\;k,\\ 0, & \mathrm{otherwise}\;(\mathrm{regarded}\;\mathrm{as}\;\mathrm{dropout}). \end{array}\right.$$

Combining (Equation 15) and (Equation 16), the probability of the successful transmission of ${\hat{x}}_{n}^{k}$ is expressed as(17)$$P\left[\lambda_{n}^{k}=1\right]=1-{\mathrm{SER}}_{n}^{k}=1-e^{-\gamma_{n}^{k}}.$$

Then, the RSE error covariance is [3](18)$$P_{n}^{k}=\left\{\begin{array}{cc} e^{\alpha }{\bar{P}}_{n}, & \mathrm{if}\;\lambda_{n}^{k}=1,\\ e^{\alpha }u_{n}\left(P_{n}^{k-1}\right), & \mathrm{if}\;\lambda_{n}^{k}=0, \end{array}\right.$$
where α is a positive parameter and $u_{n}\left(X\right)≜A_{n}XA_{n}^{T}+Q_{n}$ (see Algorithm 1).

To quantify the remote estimation quality of sensor *n* at time *k*, we define the trace of the expected RSE covariance for the sensor as follows:(19)$$\begin{array}{cc} \; & J_{n}^{k}=\mathrm{Tr}\left\{E[P_{n,r}^{k}]\right\}\\ \; & =\mathrm{Tr}\left\{P\left[\gamma_{n}^{k}=0\right]e^{\alpha }u_{n}\left(P_{n}^{k-1}\right)+P\left[\gamma_{n}^{k}=1\right]e^{\alpha }{\bar{P}}_{n}\right\} \end{array}$$

Based on the preceding system model and corresponding derivations, the optimization objective is to minimize the sum of the expected covariance of the RSE errors, which can be expressed as follows:(20)$$\begin{array}{cc} \mathrm{Problem}\;1: & \min_{\theta^{k},\;p_{n}^{k},\;b_{n,g}^{k}}\frac{1}{K}\sum_{k=1}^{K}\sum_{n=1}^{N}J_{n}^{k} \end{array}$$
(20a)$$\begin{array}{cc} s.t. & p_{n}^{k}\leq p_{\max},\;n\in N, \end{array}$$
(20b)$$\begin{array}{cc} \; & b_{n,g}^{k}\in \left\{0,1\right\},\;n\in N,\;g\in G, \end{array}$$
(20c)$$\begin{array}{cc} \; & \sum_{g=1}^{G}b_{n,g}^{k}\leq I,\;n\in N, \end{array}$$
(20d)$$\begin{array}{cc} \; & \sum_{n=1}^{N}p_{n}^{k}∥\Theta_{k}h_{n,k}^{sr}{∥}^{2}+{∥\Theta_{k}∥}^{2}\sigma_{1,k}^{2}\leq P_{\mathrm{aRIS}}. \end{array}$$
where *K* is the maximum value of the time slot set. $P_{\mathrm{aPIS}}=\xi \left(P_{\mathrm{total}}-M\left(P_{\mathrm{DC}}+P_{c}\right)\right)$ represents the amplification power of the RIS [2]. Here, ξ is the amplifier efficiency, $P_{\mathrm{total}}$ is the total power budget, $P_{\mathrm{DC}}$ is the DC biasing power consumption, and $P_{c}$ is the circuit power consumption of the active RIS. Constraint ([Disp-formula FD20a-sensors-25-04878]) ensures that the transmission power of sensor *n* is within the maximum allowed limit. Constraints ([Disp-formula FD20b-sensors-25-04878]) and ([Disp-formula FD20c-sensors-25-04878]) enforce that each channel supports at most *I* sensors and that each sensor is assigned to only one channel. These two constraints together imply that the total number of sensors *N* must not exceed the overall channel capacity, i.e., G×I≥N, in order to ensure the feasibility of the sensor-to-channel assignment. For example, if N=15 sensors are to be scheduled using G=3 channels, and each channel can support up to I=5 sensors, then the total capacity is G×I=15, and the assignment is feasible. However, if N=16, at least one sensor cannot be accommodated, violating the constraints. Therefore, this relationship is essential to guarantee a valid channel allocation. Constraint ([Disp-formula FD20d-sensors-25-04878]) limits the amplification power of the active RIS.
**Algorithm 1:** Iterative Algorithm Based on the KF Method.**Input:** $A_{n},C_{n},Q_{n},R_{n},w_{n}^{k},v_{n}^{k},\epsilon ,k_{\max}$.**Output:** $P_{n}^{k}$.**Initialization:** t=1, Δ=1.1:**while **Δ>ϵ** or **$t\leq t_{\max}$** do**2:   Calculate $P_{i,g}^{t}$ according to formulas (5)–(9)3:   Update $\Delta =P_{n}^{k|k}-P_{n}^{k|k-1}$4:   Update k=k+15:**end while**6:Obtain $P_{n}^{k}={\bar{P}}_{n}^{k|k}$7:**for** k=1 to *K* **do**8:   **for** n=1 to *N* **do**9:     **for** g=1 to *G* **do**10:        Set $b_{n,g}^{k}\in \left\{0,1\right\}$11:        **if** $b_{n,g}^{k}=1$ **then**12:          $P_{n}^{k}={\bar{P}}_{n}^{k|k}$13:        **else**14:          $P_{n}^{k}=A_{n}\left(P_{n}^{k-1}\right){\left(A_{n}\right)}^{\prime }+Q_{n}$15:        **end if**16:     **end for**17:   **end for**18:**end for**

Based on the well-known properties of the standard KF (see Lemma 2.3 in [17]), it follows that $u_{n}\left(P_{n}^{k-1}\right)\geq {\bar{P}}_{n}$, where ${\bar{P}}_{n}$ is a fixed constant. Consequently, by applying the natural logarithm function to (Equation 19), the expression can be expressed as(21)$$D_{n}^{k}=\ln\left(J_{n}^{k}\right)\approx \mathrm{Tr}\left\{-\gamma_{n}^{k}+\alpha \ln\left[u_{n}\left(P_{n}^{k-1}\right)-{\bar{P}}_{n}\right]\right\}.$$

Therefore, Problem 1 can be transformed into(22)$$\begin{array}{cc} \mathrm{Problem}\;2: & \min_{\theta^{k},p_{n}^{k},b_{n,g}^{k}}\frac{1}{K}\sum_{k=1}^{K}\sum_{n=1}^{N}(D_{n}^{k}),s.t.\;(20a),\;(20b),\;(20c),\;\left(20d\right). \end{array}$$

## 3. MADDPG-Based Resource Allocation

In this section, we first reformulate the optimization problem proposed in Problem 2 as a Multi-Agent Markov Decision Process (MDP) and then solve it using the MADDPG, Algorithm 2. As shown in Figure 2, the proposed system architecture follows a centralized training and decentralized execution (CTDE) framework, which is well suited to distributed wireless sensor networks. Specifically, the MADDPG algorithm is trained offline at a fusion center or an edge server, where both the actor and critic networks are optimized based on collected environmental data. After training, only the lightweight actor networks are deployed to individual sensor nodes, enabling each agent to independently make decisions based on its local observation without the need for online coordination. This ensures scalability and practicality in decentralized scenarios. Regarding the deployment of an active RIS, we assume it is installed at a fixed infrastructure location (e.g., a gateway node or controller hub), where an external or a battery-based power supply is available. Each active element is equipped with a low-power amplifier, and the number of elements is limited to ensure that the overall power consumption remains within practical bounds. The actions related to RIS beamforming are optimized jointly with the power control and sensor grouping through learned policies.

These implementation considerations ensure that the proposed active RIS-assisted MADDPG architecture is both theoretically grounded and practically feasible for deployment in low-power, distributed WSN environments.

**Figure 2 sensors-25-04878-f002:**
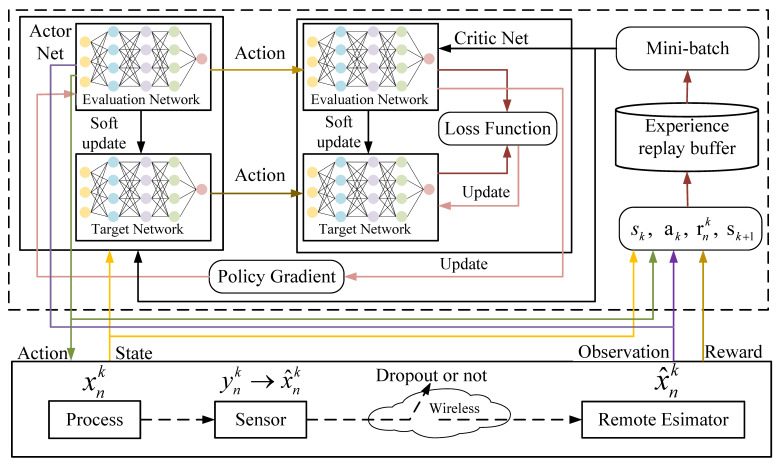
Schematic of MADDPG.

### 3.1. MDP Formulation

As depicted in Figure 2, each sensor functions as an individual agent. At time step *k*, agent *n* receives an observation $o_{n}^{k}$ from the environment and takes an action $a_{n}^{k}$. At time step *k*, given the set of states $s^{k}=\left\{o_{1}^{k},o_{2}^{k},\ldots ,o_{N}^{k}\right\}$ and the set of joint actions $a_{k}=\left\{a_{1}^{k},a_{2}^{k},\ldots ,a_{N}^{k}\right\}$, the agent receives an immediate reward $r_{n}^{k}$ and the subsequent observation $o_{n}^{k+1}$. For sensor *n*, the observable information at time *k* comprises both the channel quality state based on its current location and the remote state estimation (RSE) error covariance. This can be represented as $o_{n}^{k}=\left\{h_{n}^{k},P_{n}^{k}\right\}$. The system state at time *k* is the collection of all sensor observations, expressed as $s_{n}^{k}=\left\{o_{n}^{k}|n\in N\right\}$. At each time step, each sensor selects its action, which consists of the power allocation $p_{n}^{k}$, sensor grouping $b_{n,g}^{k}$, and beamforming selection for the RIS, denoted as $\Theta^{k}$. Thus, the action $a_{n}^{k}$ can be written as $a_{n}^{k}=\left\{\Theta^{k},p_{n}^{k},b_{n,g}^{k}\right\}$. The reward function assesses the agent’s performance and guides its decision-making. The objective of this letter is to minimize the total RSE error of all sensors. MADDPG is typically used to solve maximization problems, where each agent aims to maximize its reward by selecting actions [18]. However, Problem 2 is a minimization problem, so we need to invert the objective value by defining the reward as the negative of the cost, i.e., $r_{n}^{k}=-D_{n}^{k}=\gamma_{n}^{k}-\alpha \ln\left[u_{n}\left(P_{n}^{k-1}\right)-{\bar{P}}_{n}\right]$.

### 3.2. The MADDPG Algorithm

In this section, we propose the RIS-MADDPG algorithm to address the optimization problem outlined above. The MADDPG algorithm is a widely used multi-agent reinforcement learning (MADRL) method for continuous control problems. For *N* agents, the MADDPG algorithm involves *N* actor networks and *N* critic networks, as shown in Figure 2. Each agent’s actor network generates actions based on its policy, denoted as $\mu =\{\mu_{1},\mu_{2},\ldots ,\mu_{N}\}$, with each policy implemented by a neural network $\phi^{\mu }=\left\{\phi_{1}^{\mu },\phi_{2}^{\mu },\ldots ,\phi_{N}^{\mu }\right\}$. The critic networks, parameterized by $\phi^{Q}=\left\{\phi_{1}^{Q},\phi_{2}^{Q},\ldots ,\phi_{N}^{Q}\right\}$, evaluate the quality of the actions taken by the agents, considering the system state and the joint actions. Each agent *n* has an experience replay memory $B_{n}$ that stores transition tuples $\left(s_{k},a_{k},r_{n}^{k},s_{k+1}\right)$. At each time step, each agent samples a minibatch from its memory to update its policy. The gradient for agent *n* is computed as follows [18]:(23)$$\begin{array}{cc} \; & \nabla_{\phi_{n}^{\mu }}J\left(\mu_{n}\right)=\\ \; & E_{a_{n}^{k}∼B_{n}}\left[\nabla_{\phi_{k}^{\mu }}\mu_{n}\left(a_{n}^{k}|o_{n}^{k}\right)Q_{n}^{\mu }\left(s^{k},a_{k}\right){|}_{a_{n}^{k}=\mu_{n}\left(o_{n}^{k}\right)}\right]. \end{array}$$

To stabilize the learning process, MADDPG employs the target network technique. The target networks for the action and evaluation functions, $\mu_{n}^{\prime }$ and $Q_{n}^{\prime }$, respectively, are updated using backpropagation. The loss function for updating the evaluation network is(24)$$L\left(\phi_{n}^{Q}\right)=E_{s^{k},a_{k},r_{n}^{k},s_{k+1}^{\prime }}\left[{\left(Q_{n}^{\mu }\left(s^{k},a_{k}\right)-y_{k}\right)}^{2}\right],$$
where $y_{k}=r_{n}^{k}+\psi Q_{n}^{\prime }\left({\left(s^{k}\right)}^{\prime },a_{k}^{\prime }\right){|}_{{\left(a_{n}^{k}\right)}^{\prime }=\mu_{n}^{\prime }\left(o_{n}^{k}\right)}$.

After updating $\phi_{n}^{\mu }$ and $\phi_{n}^{Q}$, the parameters of the target networks, $\phi_{n}^{\mu^{\prime }}$ and $\phi_{n}^{Q^{\prime }}$, are softly updated as(25)$$\begin{array}{cc} \; & \phi_{n}^{\mu^{\prime }}=\tau_{a}\phi_{n}^{\mu }+\left(1-\tau_{a}\right)\phi_{n}^{\mu^{\prime }}, \end{array}$$(26)$$\begin{array}{cc} \; & \phi_{n}^{Q^{\prime }}=\tau_{c}\phi_{n}^{Q}+\left(1-\tau_{c}\right)\phi_{n}^{Q^{\prime }}, \end{array}$$
where $\tau_{a}$ and $\tau_{c}$ are the update parameters for the target network, and C denotes the update interval that determines how frequently the actor and target networks are updated during training. The detailed steps of the RIS-MADDPG algorithm are provided in Algorithm 2.

**Algorithm 2** RIS-MADDPG Algorithm.
1:Initialize the actor network $\mu_{n}\left(o_{n}^{k};\theta_{n}^{\mu }\right)$ and the critic network $Q_{n}\left(s_{n}^{k};\theta_{n}^{Q}\right)$ for each agent *n* at time step *k*;2:Initialize the target network $\mu_{n}^{\prime }\left(o_{n}^{k};\theta_{n}^{\mu^{\prime }}\right)$ and the critic nework $Q_{n}^{\prime }\left(s_{n}^{k};\theta_{n}^{Q^{\prime }}\right)$ for each agent *n* at time step *k*;3:Initialize the experience replay buffer $B_{n}$ for each agent *n*;4:**for** episode = 1 to E **do**5:   Initialize state $s^{k}=\left\{o_{1}^{k},o_{2}^{k},\ldots ,o_{N}^{k}\right\}$ and k=16:   **while** step k<K **do**7:     **for** each agent *n* **do**8:        Get $o_{n}^{k}$, select action $a_{n}^{k}=\mu_{n}\left(o_{n}^{k}\right)$9:        Execute actions $a_{k}$ and obtain the reward $r_{n}^{k}$ and the next state $o_{n}^{k+1}$10:        Store $\left(s_{k},a_{k},r_{n}^{k},s_{k+1}\right)$ in $B_{n}$11:     **end for**12:   **end while**13:   **for** each agent *n* **do**14:     Sample a random mini-batch of S samples from the experience replay buffer $B_{n}$15:     Update the critic network by minimizing the loss function $L(\phi_{n}^{Q})$ in (Equation 24)16:     **if** kmodC=0then17:         Update the parameters $\phi_{n}^{Q}$ using the deterministic policy gradient in (Equation 23)18:         Update two target networks: (Equation 25) and (Equation 26)19:   **end if**20: **end for**21:
**end for**



To ensure the reproducibility of our results, all of the hyperparameters, training configurations, and simulation environment details have been clearly specified in Table 1, and the algorithmic process has been fully described in Algorithm 2.

**Table 1 sensors-25-04878-t001:** Simulation parameters.

Description	Parameter and Value
Transmission model	N = 9, G = I = 3, M = 16, *p*_max_ = 20 dBm
	*δ_R_* = *δ_R_* = 4 *P*_aRIS_ = 32 dBm
	*P*_tot_ = 30 dBm, *P*_DC_ = −5 dBm, *P_c_* = −10 dBm
	*α_R_* = 2.7, *α_B_* = 1.5, *α_S_* = 1.6, *α_m_* ∈ [1.1, 2]
	*ξ* = 0.8, *α* = 10^17^, $\sigma_{1}^{2}$ = −70 dBm, $\sigma_{2}^{2}$ = −80 dBm
KF model	**A***_n_*, **C***_n_*, **Q***_n_*, **R***_n_* ∈ [0.5, 1.5]
>MADDPG	>E = 10, K = 4000, ℬ_*n*_ = 128, C = 2000

### 3.3. Computational Complexity Analysis

The computational complexity of Algorithm 1 mainly arises from two iterative phases: the convergence phase for updating the estimation covariance $P_{n}^{k|k}$ and the scheduling phase involving nested loops over time slots (*K*), agents (*N*), and groups (*G*). Specifically, the convergence phase’s complexity is $O(k_{\max}\times N\times G)$, and the scheduling phase’s complexity is O(K×N×G). Thus, the total complexity of Algorithm 1 is $O(N\times G\left(k_{\max}+K\right))$, reflecting linear scalability in terms of the agents and groups.

For the MADDPG algorithm used to solve Problem 2, the complexity consists of two parts: centralized critic network training and decentralized actor network execution. The centralized training’s complexity is $O(E^{\max}\times N)$, where $E^{\max}$ is the maximum number of training episodes. The decentralized execution complexity per agent is constant per episode, resulting in an overall execution complexity of $O(E^{\max})$. Hence, the total complexity is $O(E^{\max}\times N)$. Compared to centralized approaches like DDPG (with complexity $O(E^{\max}\times N^{2})$), MADDPG significantly reduces the computational overhead, thus enhancing the scalability and efficiency for multi-agent scenarios.

## 4. Simulation Results

Based on the above discussion, we divide the computational complexity of applying the MADDPG algorithm to solving Problem 2 into three components:

The critic network (centralized training): During the centralized training phase, the critic network evaluates the global state–action pairs involving all *N* agents. Suppose that the maximum number of episodes is denoted as $E^{\max}$, and each agent requires a complexity of O(1) for a single update: the overall complexity becomes $O(E^{\max}N)$.The actor network (decentralized execution): In the decentralized execution phase, each agent independently selects actions using its actor network. Since each agent makes decisions independently with the complexity O(1) per episode, the total complexity for all episodes is $O(E^{\max})$.

Therefore, the total computational complexity of the MADDPG algorithm is represented as(27)$$O\left(E^{\max}N+E^{\max}\right)=O\left(E^{\max}N\right)$$

Moreover, we critically analyze the overhead costs compared to those of the baseline models. Specifically, compared to the centralized DDPG method—which incurs a complexity of approximately $O(E^{\max}N^{2})$ due to the joint optimization of the actions over all agents—MADDPG significantly reduces the complexity by decentralizing the actor network computations, compared to that of the standard reinforcement learning methods.

Figure 3 compares our proposed MADDPG algorithm with active RIS assistance against three baseline approaches: MADDPG with a passive RIS, DDPG with an active RIS, and MADDPG without RIS assistance. The results demonstrate substantial reductions in the performance errors provided by the active RIS. Specifically, the MADDPG algorithm with active RIS assistance achieves an 88% reduction in error compared to that with a passive RIS and a 200% reduction compared to that with no RIS assistance. Additionally, the active RIS within the MADDPG framework surpasses active-RIS-assisted DDPG by approximately 87% in its error reduction. These results clearly highlight the superior performance of the MADDPG algorithm with the active RIS, underscoring the potential of the active RIS to optimize WSNs. To enhance the clarity further, we have specified in the figure captions that the horizontal axis denotes the iteration steps and the vertical axis corresponds to the normalized estimation errors, both of which are dimensionless quantities.

Figure 4 illustrates the impact of the number of reflection elements on the sensor’s RSE error. It can be observed that increasing the number of elements to 128 results in substantial error reductions of 32%, 78%, and 120% when compared to configurations with 32, 16, and 8 elements, respectively. This performance gain is attributed to the improved signal reflection and enhanced spatial diversity enabled by more reflection elements. These results demonstrate that a higher number of elements significantly enhances the sensing accuracy, thereby playing a crucial role in reducing the estimation error.

Figure 5 presents the convergence behavior of the RSE error under different maximum power constraints (10 dBm, 15 dBm, and 20 dBm). While the general convergence trends are visually similar, higher power constraints clearly lead to faster convergence and a lower steady-state RSE error. This is because larger transmission power budgets improve the signal-to-noise ratio (SNR) at the receiver, enabling more precise state estimations. These findings indicate that the power allocation not only influences the final estimation accuracy but also affects the convergence speed, highlighting its critical importance in dynamic WSN environments. In summary, the results in Figure 4 and Figure 5 jointly confirm that both the number of reflection elements and the power constraints have a pronounced impact on the sensing performance. Optimizing these parameters leads to more accurate and responsive sensing, which is essential for reliable state estimation in resource-constrained wireless sensor networks.

**Figure 3 sensors-25-04878-f003:**
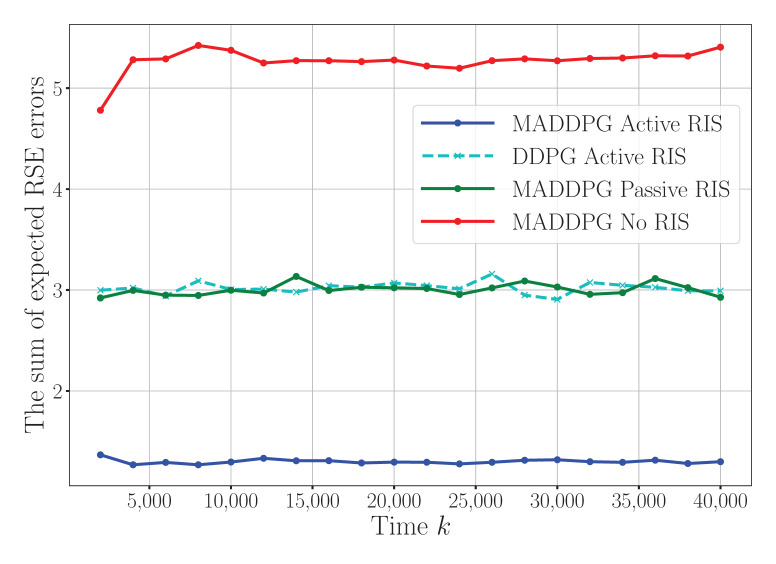
The sum of expected RSE errors under different algorithms.

**Figure 4 sensors-25-04878-f004:**
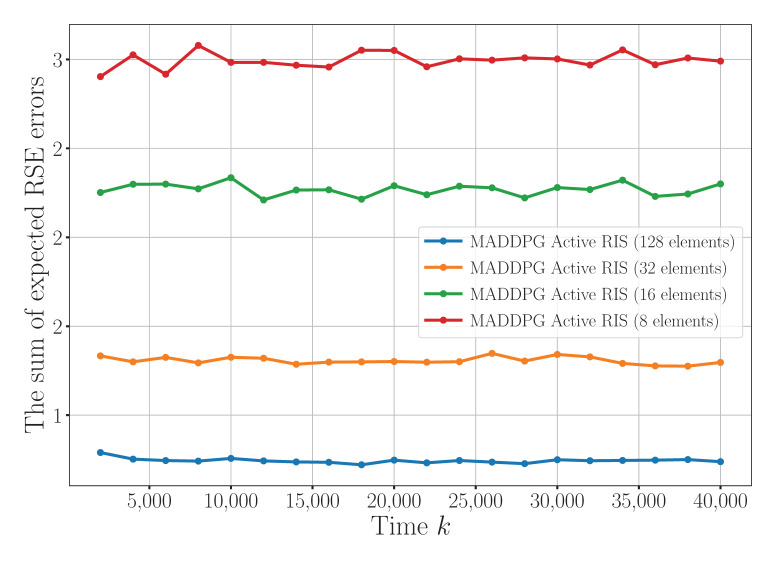
The convergence for different numbers of RIS reflecting elements.

**Figure 5 sensors-25-04878-f005:**
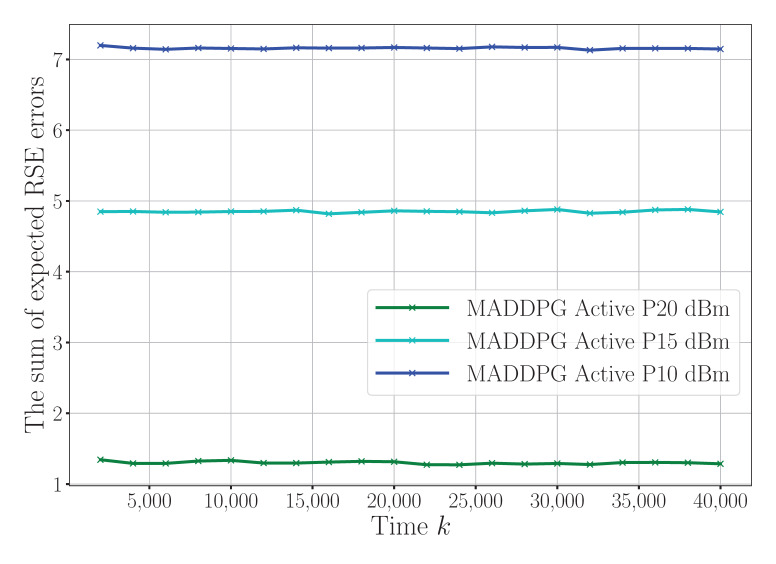
Convergence for different maximum power constraints.

To investigate the system-level trade-offs further, Figure 6 presents the convergence behavior of the proposed algorithm under different sensor-to-channel configurations (e.g., two sensors with six channels, three sensors with four channels, and four sensors with three channels). It can be observed that as the number of sensors increases under limited channel resources, the RSE error grows due to heightened interference and limited degrees of freedom for resource allocation. This validates the system’s sensitivity to the sensor density and available spectrum.

**Figure 6 sensors-25-04878-f006:**
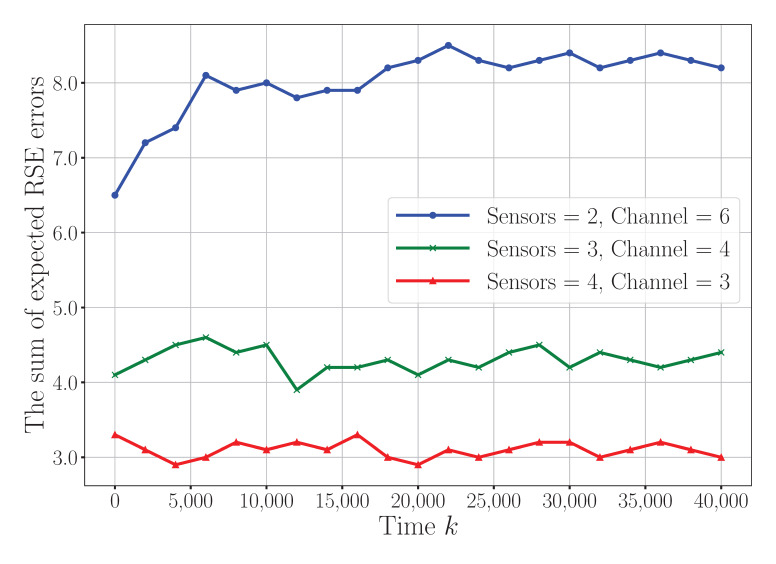
Convergence comparison under different sensor-to-channel configurations.

Figure 7 evaluates the impact of different amplifier efficiency coefficients ξ on the convergence behavior of the proposed algorithm. The parameter ξ reflects the effective amplification capability of the active RIS, as constrained by the total power budget $P_{\mathrm{total}}$ and the internal power consumption $M(P_{\mathrm{DC}}+P_{c})$, as defined in constraint (20d). A larger value for ξ enables the active RIS to contribute more effectively to signal enhancement via stronger amplification, thereby improving the signal-to-noise ratio (SNR) at the receiver and reducing the RSE error. However, increasing ξ also implies that more power is allocated to RIS amplification, potentially limiting the power available for sensors. As shown in the figure, a moderate value for ξ achieves a good trade-off between RIS enhancement and the overall system power balance, whereas excessive amplification may lead to diminishing returns. This result highlights the importance of careful tuning of the RIS-related parameters to optimize the joint communication–sensing performance in energy-constrained WSN environments.

**Figure 7 sensors-25-04878-f007:**
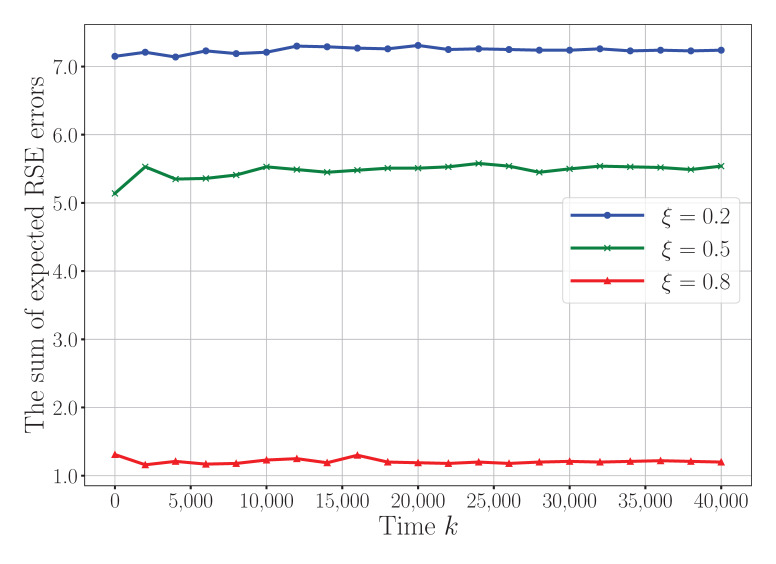
Convergence performance under different RIS amplifier efficiency coefficients ξ.

To facilitate a comprehensive and intuitive comparison across different experimental settings, Table 2 presents a summary of the convergence time and the steady-state RSE error under various configurations. In this table, “Conv. Time” denotes the number of iterations (abbreviated as “Iter.”) required for the algorithm to reach convergence, while “Conv. Value” represents the final RSE error after stabilization. The data in the table are systematically derived from key experimental results, primarily based on the quantitative trends observed in Figure 3, Figure 4 and Figure 5. As shown in Table 2, the proposed MADDPG algorithm assisted by the active RIS consistently achieves a superior performance in terms of both the convergence speed and estimation accuracy, particularly when equipped with a larger number of reflecting elements. In contrast, reducing the number of RIS elements or the transmission power constraint leads to increased estimation errors, highlighting the importance of spatial diversity and power resources for accurate sensing. This table provides a clear and concise overview of the system behavior across algorithms and configurations, offering valuable insights for future research and parameter optimization in wireless sensor networks.

**Table 2 sensors-25-04878-t002:** A summary of the convergence times and convergence values under different configurations.

Scenario	Conv. Time (Iter.)	Conv. Value
MADDPG + Active RIS (128 elements, 20 dBm)	3500	0.75
MADDPG + Active RIS (32 elements, 20 dBm)	3300	1.25–1.35
MADDPG + Active RIS (16 elements, 20 dBm)	3200	2.20–2.35
MADDPG + Active RIS (8 elements, 20 dBm)	3000	3.00
MADDPG + Active RIS (16 elements, 10 dBm)	3500	7.10–7.25
MADDPG + Active RIS (16 elements, 15 dBm)	3500	4.85–5.00
DDPG + Active RIS (16 elements, 20 dBm)	4500	3.10–3.20
MADDPG + Passive RIS (16 elements, 20 dBm)	5000	3.00
MADDPG without RIS (16 elements, 20 dBm)	10,000	5.15–5.25

## 5. Conclusions

This letter proposes an active RIS-assisted uplink NOMA technique using the MADDPG framework, aiming to minimize the sum of the expected RSE errors during wireless transmission. By leveraging the capabilities of an active RIS and the multi-agent learning capability of MADDPG, the proposed method dynamically optimizes the resource allocation to enhance the sensing and transmission performance in wireless networks. The simulation results validate that the proposed algorithm achieves notable improvements in minimizing the sum of the RSE errors, outperforming the conventional algorithms by approximately 87% to 200%. These results demonstrate the effectiveness of combining intelligent reflecting surfaces with reinforcement learning for joint communication and sensing design. Despite these promising results, the proposed approach also has several limitations. The offline training phase of MADDPG can be computationally intensive, especially with many agents, and the convergence behavior may be sensitive to hyperparameter tuning. In future work, we plan to explore more lightweight and scalable learning strategies while also evaluating additional performance metrics such as the convergence speed, latency, and energy efficiency. Moreover, extending the framework to support mobile agents and distributed multi-cell deployments will enhance its applicability further in real-world WSN scenarios.

## Figures and Tables

**Figure 1 sensors-25-04878-f001:**
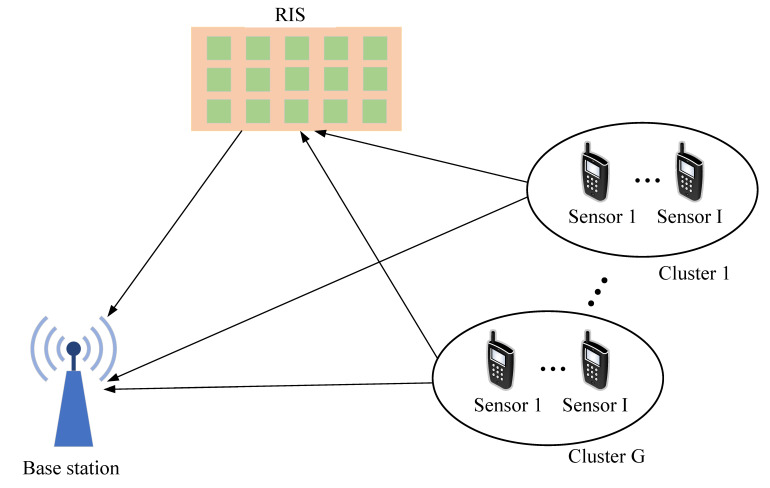
Active RIS-assisted uplink NOMA model.

## Data Availability

The data that support the findings of this study are available from the corresponding author upon reasonable request.

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
