# Peer review of "Active RIS-Assisted Uplink NOMA with MADDPG for Remote State Estimation in Wireless Sensor Networks"

_sensors, 2025, doi:10.3390/s25154878_

Round 1

Reviewer 1 Report

Comments and Suggestions for Authors

This paper proposes a joint optimization of RIS, NOMA and sensor readings. It isa well written manuscript and technically sound, but it is not finalized properly. The first sections are almost outstanding, but then the section of results is brief and lacks details. The figures of results do not actually provide great information except of the horizontal levels. The authors have to find a better way to provide a tradeoff analysis of the proposed systems with figures that are more significant than three or 4 levels of metrics.

Reviewer 2 Report

Comments and Suggestions for Authors

This study tackles computational complexity and non-convex optimization challenges in NOMA-RIS systems for beyond-5G networks. It develops a MADDPG-based framework that jointly optimizes sensor grouping, power allocation, and RIS parameters, achieving significant RSE error reduction. Simulations validate its superior performance in RIS-assisted NOMA systems.

There are some issues in this paper that need to be supplemented and improved, as described below.

  1. The phrase in Line 46 states "the system consists of a single BS, equipped with G receive antennas," while Line 49 mentions G sensor groups. Does this imply that signals from sensors within different sensor groups are received by their corresponding antennas, respectively? The system model should provide a more detailed explanation regarding the relationship between these notations and the channel configuration.
  2. The expression in Line 51, " where x_n^k Rln and y_n^k Rrn " should clearly define the notations ln and rn.
  3. Equation (2) is cited from reference [16], but differs from the equation in [16]. The reason for this discrepancy should be further clarified.
  4. The expression " posterior state estimate x ̂_n^(k-1) at time k " below equation (4) should be corrected to x ̂_n^k.
  5. The notation y_g^k in Equation (10) should clarify why both indices k and g are presented in boldface.
  6. The notation P_n^k in Line 60, denoting steady-state covariance, and p_n^k in Line 80, representing computation offloading power, may confuse readers due to their visual similarity.
  7. Equation (14) expresses the signal-to-interference-plus-noise ratio (SINR) of sensor n. The rationale for excluding sensors with indices smaller than n from the interference terms should be explicitly justified.
  8. The expression " Tr{E[P_(n,r)^k]}" in Equation (19) should explicitly define the meaning of the subscript r.
  9. The statement in Line 120, "Constraints (20b) and (20c) enforce that each channel supports at most I sensors and that each sensor is assigned to only one channel, which means G × I ≤ N," requires further elaboration for clarity. Please provide an illustrative example.
  10. The notation for sets should remain consistent throughout - for instance, boldface is used in Line 135 but regular font appears in Line 139.
  11. In this paper, the time index k is mostly placed in a superscript position. It is recommended to maintain this notational consistency for 'the set of states' and 'the set of joint actions' in Line 135 as well.
  12. In Algorithm 1, line 16, the condition 'if k mod C = 0' requires a prior formal definition of parameter C.
  13. The subsection heading '3.3. Algorithm Development' is not entirely appropriate; the authors may consider modifying or removing it.
  14. Figures 3 through 5 should include units for all axes.
  15. Why are the results in Figure 3 and Figure 4 different for MADDPG Active RIS (M = 16)?

Reviewer 3 Report

Comments and Suggestions for Authors
  1. In Figure 1, which focuses on the active RIS-assisted uplink NOMA model, the authors should check the directions of the lines with arrows; most of them are wrong. Also, the figure is poorly presented, as most of the technical information, such as the channel matrices, is missing.
  2. The key contributions of the paper should be highlighted and elaborated in section 1. Also, the paper organization should be added at the end of section 1.
  3. The related work section is limited in scope and should be strengthened. The pros and cons of existing works should be clearly stated and demonstrated, showing how the current work addresses the identified gaps.
  4. The rationale for proposing the MADDPG algorithm needs to be carefully defined, and all assumptions made in its design should be clarified.
  5. The benefits or advantages of the MADDPG algorithm over the existing algorithms should be demonstrated.
  6. It is not clear if the Algorithm 1 RIS-MADDPG Algorithm is sufficient to reproduce the results in this study.
  7. The computational complexity and overhead costs of the MADDPG Algorithm should be critically analyzed and compared to the baseline models.
  8. The performance evaluation of the results is weak and should be strengthened.
  9. It is not clear why the results in Figure 5. Convergence for different maximum power constraints, and shows limited variations as the time varies.
  10. The limitations of the proposed algorithm need to be discussed, and future scope should state how these shortcomings can be addressed.
  11. Minor English editing is required throughout the paper.
Comments on the Quality of English Language

Minor English editing is required.

Reviewer 4 Report

Comments and Suggestions for Authors

The authors propose an Active Reconfigurable Intelligent Surface (RIS)-assisted uplink Non-Orthogonal Multiple Access (NOMA) framework for remote state estimation in wireless sensor networks. They incorporate a multi-agent deep deterministic policy gradient (MADDPG) reinforcement learning method to jointly optimize RIS phase shifts and transmission strategies to improve estimation performance. I have some comments about the paper.

  1. While the use of MADDPG in this context is novel to some extent, the core idea follows existing reinforcement learning frameworks. The paper lacks theoretical performance analysis, convergence study, or complexity comparison.
  2. The authors discuss prior works on NOMA, RIS, and RL-based optimization, but the literature review is missing key developments in low-complexity signal processing and cross-domain hybrid learning schemes. In this context, the authors are encouraged to cite, Low Complexity MIMO-FBMC Sparse Channel Parameter Estimation for Industrial Big Data Communications.
  3. The feasibility of deploying active RIS in low-power, distributed sensor networks is not fully addressed. Please clarify: What is the assumed power consumption model for the active RIS elements? How realistic is the assumption of centralized coordination with MADDPG in WSNs?
  4. The simulations focus only on MSE and estimation cost. To strengthen the evaluation,pls include latency, convergence speed, or energy efficiency comparisons.

Round 2

Reviewer 1 Report

Comments and Suggestions for Authors

The authors have provided answer to the previous criticism. It is satisfactory. The suggestion of this reviewer is that the authors can also present the results in an additional format. All the figures show convergence of the results with execution time or iterations. You can haver an additional figure or figures, or even tables with a summary of all these results both in convergence time and the value of convergence. This would add a different style to the paper and provide a good overview of the other figures, whose format seem too repetitive and carrying limited information.

Reviewer 2 Report

Comments and Suggestions for Authors

No further comments.

Author Response

We sincerely thank the reviewer for their continued time and effort in evaluating our manuscript. We appreciate that no further comments were raised during this revision round. Accordingly, no additional modifications were made in this version in response to this review. We are also grateful for the reviewer’s previous suggestions, which have already helped improve the quality and clarity of the manuscript in earlier revisions.

Reviewer 3 Report

Comments and Suggestions for Authors

The authors have addressed my concerns.

Author Response

(The authors gave the same response as above.)

Reviewer 4 Report

Comments and Suggestions for Authors

The paper is well revised, i have no more comments.

Author Response

(The authors gave the same response as above.)
